# Breastfeeding Duration and Timing of Bottle Supplementation: Associations with Body Mass Index from Childhood to Young-Adulthood

**DOI:** 10.3390/nu15143121

**Published:** 2023-07-13

**Authors:** Estela Blanco, Suzanna M. Martinez, Patricia East, Raquel Burrows, Paulina Correa-Burrows, Betsy Lozoff, Sheila Gahagan

**Affiliations:** 1Centro de Investigación en Sociedad y Salud y Núcleo Milenio de Sociomedicina, Las Condes, Santiago 7550000, Chile; 2Department of Epidemiology and Biostatistics, University of California, San Francisco, CA 90095, USA; suzanna.martinez@ucsf.edu; 3Department of Pediatrics, Division of Child Development and Community Health, University of California, San Diego, CA 92093, USA; peast@health.ucsd.edu (P.E.); sgahagan@health.ucsd.edu (S.G.); 4Institute of Nutrition and Food Technology, University of Chile, Macul, Santiago 7810000, Chile; rburrows@inta.uchile.cl (R.B.); paulina.correa@inta.uchile.cl (P.C.-B.); 5Department of Pediatrics, University of Michigan, Ann Arbor, MI 48109, USA; blozoff@umich.edu

**Keywords:** breastfeeding, body mass index, life course

## Abstract

Evidence for the association between breastfeeding (BF) duration and later body mass index (BMI) is inconsistent. We explored how BF duration and BF type (exclusive or partial) related to BMI from childhood to young adulthood in a Chilean cohort. Infants were recruited at 6 months between 1994 and 1996 in Santiago, Chile (*n* = 821). Mothers reported date of first bottle and last BF; anthropometry was measured at 1, 5, 10, 16, and 23 years. We tested whether: (1) type of BF at 6 months (none, partial, exclusive) and (2) duration of exclusive BF (<1 month, 1 to <3 months, 3 to <6 months, and ≥6 months) related to BMI. At 6 months, 35% received both breastmilk and formula (“partial BF”) and 38% were exclusively breastfed. We found some evidence of an association between longer BF and lower BMI z-scores at young ages but observed null effects for later BMI. Specifically, BF for 3 to <6 months compared to <1 month related to lower BMI z-scores at 1 and 5 years (both *p* < 0.05). Our results are in partial accordance with others who have not found a protective effect of longer BF for lower BMI.

## 1. Introduction

The conditions of overweight and obesity relate to increased risk of a number of negative health outcomes, including cardiovascular diseases (e.g., heart disease and stroke), diabetes, and some cancers (e.g., breast, ovarian, prostate, and liver, among others) [1]. In 2016, it was estimated that 39% of adults globally were overweight (body mass index [BMI] ≥ 25) or obese (BMI ≥ 30), with important differences between countries based on income and level of development [1,2]. For example, in the United States, almost 70% of adults are overweight or obese, while the prevalence is closer to 20% in South Asia [2]. Childhood overweight and obesity is particularly worrying, considering that weight status often tracks into adulthood. Similar to time trends among adults, childhood overweight and obesity has increased at an astonishing rate in the past 40 years. The World Health Organization (WHO) reports that the prevalence of overweight (weight-for-height > 2 standard deviations above growth standards) and obesity among children and adolescents (aged 5–19) has increased from 4% in 1975 to 18% in 2016, with similar increases for boys and girls [1]. For obesity (weight-for-height > 3 standard deviations above growth standards), in 1975, only 1% of the pediatric population was considered obese, while in 2016 the prevalence was around 7% [1]. These alarming statistics highlight the need for evidence on effective interventions for overweight and obesity.

There is a growing body of evidence that markers for increased risk for later cardiometabolic health, including obesity, begin in early life. Early life nutrition and its associated factors represent potentially important points of intervention for lifelong cardiometabolic benefits. Breastfeeding (BF) is an example of an early life nutritional factor that is related to increased risk of obesity and blood pressure in childhood in some, but not all, studies. The WHO recommends breast milk as the sole source of nutrition for infants up to 6 months (m) of age and continued BF with complementary food from 6 m to 2 years [1]. There is strong evidence of the benefits of BF for infant and mother in both the short- and long-term. In the short term, BF is associated with lower risk of common childhood infections (e.g., otitis and diarrhea) and the mother’s feelings of greater attachment and closeness to her baby [2]. In the long term, for the child, BF is associated with a lower risk of asthma (independent of family history of asthma) and type 2 diabetes; and for the mother, a lower risk of breast and ovarian cancer [2].

The relationship between BF and later BMI and obesity risk remains controversial. Some studies, mostly from developed countries, find a protective relationship (i.e., longer BF associated with lower BMI or lower risk of obesity) [3,4,5], while others from developing countries do not [6,7]. For example, a study conducted in Australia with follow-up at 20 years found that infants with <6 m BF had about a 1-point higher BMI at 17 years, compared to those who received breastmilk for >6 m [4]. However, another study conducted in the Pelotas region of Southern Brazil found that BF was unrelated to BMI at 11 years [6]. Thus, the question remains as to whether BF is causally associated with later BMI or whether the relation is sometimes observed due to residual confounding. In general, duration of exclusive BF (breastmilk as the sole source of milk) is longer in low- and middle-income countries than in high-income countries [8]. Within both developed and developing countries there is often a socioeconomic status (SES) divide such that women with higher levels of education generally BF longer than women with less education [9,10]. Given the potential for confounding both between and within countries based on SES, it is important to continue to evaluate the relation between BF and BMI in countries of various levels of SES.

Comparisons between studies are also difficult given that BF is often defined differently. For example, some studies specify exclusive lactation whereas others do not, and some categorize duration of lactation differently (e.g., as ≥3, 6, 7 m or total BF). Other studies compare exclusive BF to no BF: comparing infants who never received bottle supplementation to infants who were never breastfed (i.e., exclusively bottle fed), which is likely appropriate in contexts where BF is uncommon. The current study took place in Chile, a country with high rates of BF. Recent nationally representative data reveal that most Chilean infants (80%) are exclusively breastfed for 1 m and 56% for 6 m [11]. Furthermore, comparing exclusive BF to exclusive bottle feeding (no BF) essentially ignores the potential effects of partial or mixed feeding (i.e., bottle feeding with infant formula while continuing to BF). Understanding the potential health benefits of a variety of BF experiences is important given that many women struggle to meet WHO recommendations and begin bottle supplementation with or without continued BF before 6 m [12,13].

The objective of the current analysis was to evaluate the relation between BF and BMI as measured in childhood, adolescence, and adulthood, using a birth cohort from Santiago, Chile. Chile is a particularly interesting context to study this research question, as it has undergone rapid development beginning in the 1990s [14,15] and has high rates of BF. Furthermore, having detailed information of BF duration and introduction of first bottle of supplemental milk collected in the first year of life allows for exploration of the effect of partial BF and duration of exclusive BF. We hypothesized that exclusive and partial BF at 6 m, compared to no BF and longer exclusive BF (>1 m), would relate to a lower BMI throughout childhood and young adulthood among Chilean children born between 1994 and 1996. The findings of our study may help to clarify the potential protective effect that longer duration of BF may have in relation to lower BMI in childhood, adolescence, and young adulthood in a context of rapid economic development and high BF rates.

## 2. Materials and Methods

The current study involved participants who were recruited from the second half of an infancy study (1994–1996) of iron deficiency anemia (IDA) when there was no minimum intake of formula required (*n* = 821) and followed up at 5 years (*n* = 538), 10 (*n* = 534), 16 (*n* = 508, mean age 16.2 y) and 23 years (*n* = 511, mean age 23 y). Infants were enrolled in either a preventive trial of IDA [16] or a neuromaturation study [17] and recruited from community clinics in four low- and middle-income neighborhoods of Santiago, Chile. Inclusion criteria included: singleton and uncomplicated vaginal birth at term (>37 weeks’ gestation) with birthweight ≥ 3 kg, no major congenital abnormality, and no prior iron therapy. Infants without IDA entered the preventive trial, while IDA infants, along with the next nonanemic control, entered the neuromaturation study. For the preventive trial, infants were randomly allocated to either an iron supplementation or usual nutrition group. Infants in the neuromaturation study received medicinal iron. All participants were recruited around 6 m of age and followed intensively until 1 year of age for the infancy iron study [16]. Follow-ups were conducted at 5 years in a subset of the original cohort, due to a substantial budget cut, and the full cohort was invited to participate in follow-up evaluations roughly every 5–7 years thereafter, starting at age 10. Institutional Review Boards at the participating universities approved this study in accordance with the Declaration of Helsinki. Written informed consent was provided prior to study enrollment. Specifically, parental consent was obtained in infancy, 5, 10, and 16 years. Participants provided assent at 10 and 16 years and informed consent at 23 years.

Data collection. Between 6 and 12 m, infants were measured (weight and length) monthly in community clinics, and the primary caretaker completed evaluations that included basic demographic information (e.g., highest level of education and age). Medical records were used to obtain information prior to 6 m, including gestational age, birth weight and sex at birth. At all subsequent follow-ups, participants were weighed and measured at the Institute of Nutrition and Food Technology (INTA), University of Chile, using standardized procedures.

Exposure. For infants receiving formula/cow milk supplementation at 6 m, mothers retrospectively reported date of first bottle. For all other infants, mothers were asked weekly, between 6 and 12 m, if they continued to BF and if infants had received bottle supplementation. Because expressed human milk feeding by bottle was rare in Chile, date of first bottle was considered equivalent to the end of BF as the sole source of milk [18]. Initiation of BF was almost universal in this sample, with only 3 participants never breastfed. We studied BF as type of BF at 6 m and duration of exclusive BF. Type of BF at 6 m was categorized as: none (exclusively bottle fed, date of first bottle and date of last BF < 180 days), partial (both BF and bottle fed, date of first bottle < 180 and date of last BF ≥ 180), and exclusive (no bottle feeding, date of first bottle and date of last BF ≥ 180). Duration of exclusive BF was studied as having a date of first bottle <1 m, 1 to <3 m, 3 to <6 m, and ≥6 m. For both BF variables, the lowest BF group (no BF at 6 m; exclusive BF < 1 m) was chosen as the reference for ease of comparisons with other studies [4,6].

Outcome. Body mass index (BMI) z-score was calculated based on WHO criteria [19] and used as the outcome at ages 1, 5, 10, and 16 years. At 23 years, BMI (kg/m^2^) was used.

Statistical analysis. Variables were described with mean (standard deviation) or frequency, overall and by BF group. To compare means of BMI by BF group, we used analysis of variance (ANOVA). We tested two linear regression models at age 1, 5, 16, and 23 years. Model 1 examined type of BF at 6 m (none, partial, or exclusive) with BMI and Model 2 examined duration of exclusive BF with BMI. Both models adjusted for the following covariates: child sex, maternal education, maternal age, gestational age, and birth weight. We also tested for a linear trend of BF category. All analyses were conducted in Statistical Package for the Social Sciences (SPSS) and a *p*-value of <0.05 indicated statistical significance.

## 3. Results

Infants weighed on average 3.58 kg (standard deviation (SD) = 0.3) at birth, with slightly more males (53%) than females in the cohort. Mothers were on average 26.5 (SD = 6.1) years of age, with 9.5 (SD = 2.6) years of education. With respect to type of BF at 6 m, 26% had no BF, 34% partial BF, and 38% exclusive BF. For duration of exclusive BF, most were breastfed for 3–6 (30%) or 6 m or greater (38%), with approximately equal proportions for lower durations of exclusive BF (16% in both <1 m and 1–3 m groups). Additional details can be found in Table 1.

Table 2 shows mean (SD) BMI at all ages evaluated by BF group (type of BF at 6 m and duration of exclusive BF). No differences shown in Table 2 were statistically significant (all *p* > 0.05), however some patterns could be observed. Partially BF infants (those who received bottle supplementation along with BF at 6 m) consistently had lower average BMI (at 1, 5, 10, 16, and 23 years) compared to the no BF group, although some differences were nominal (26.3 versus 27.2 at 23 years). Additionally, the exclusive BF group had a slightly lower BMI z-score than those with no BF at 1 year (0.84 compared to 0.91), but a higher BMI z-score later in life (BMI z-score at 16 y of 1.05 versus 0.83). For duration of BF, infants who received supplementary milk for the first time between 3 and 6 m of age had the lowest BMI at all time points compared to the other duration groups, but differences were modest especially at later ages. We observed no clear patterning of BMI by longer duration of BF.

Table 3 shows that, compared to the no BF group, the partial BF (mixed BF and bottle feeding) and exclusive BF groups had lower BMI z-score at ages 1 and 5 years, after adjusting for covariates. However, the relations reached statistical significance at age 5 for the partial BF group only (Table 3). Specifically, infants who were partially BF at 6 m had 0.31 lower BMI z-score (95% confidence interval: −0.57, −0.05) at 5 y, compared to infants who were not BF at 6 m. With respect to duration of exclusive BF, a clear trend was not observed. However, we note that more BF appeared to relate to lower BMI at all time points, although most comparisons contained the null value. We found that compared to <1 m of exclusive BF, exclusive BF for between 3 and 6 m was associated with lower BMI at all time points and statistically significant at 1 and 5 years. Infants who were BF exclusively for between 3 and 6 m (3 to <6 m group), compared to those who received < 1 m exclusive BF, had a 0.22 (95% confidence interval: −0.41, −0.00) and 0.31 (95% confidence interval: −0.61, −0.01) lower BMI z-score at 1 and 5 years, respectively. We found no evidence of a linear relationship between type of BF at 6 m or duration of exclusive BF and BMI at any age evaluated.

## 4. Discussion

In our cohort of Chilean infants born in the 1990s and followed up in childhood, adolescence, and young adulthood, we observed limited evidence of an inverse relationship between BF (type at 6 m and duration of exclusive BF) and lower BMI later in life. We hypothesized that more BF (partial and exclusive BF at 6 m) and longer exclusive BF (>1 m) would relate to lower BMI at later ages. Study findings indicated some seemingly protective effects of more and longer BF for lower BMI at ages 1 and 5. Specifically, partial compared to no BF at 6 m related to a lower BMI z-score at 5 years and exclusive BF for 3–6 m, compared to <1 m, related to lower BMI z-scores at 1 and 5 years. We note that while statistically significant, the magnitude of these effects was small and, thus, clinical significance is likely limited. However, results did not support a clear linear dose–response relation and there were no significant effects for exclusive BF at 6 m or ≥6 m duration. Additionally, no significant BF effects were found at later childhood, adolescence, or early adulthood.

With respect to duration of BF, our results are similar to those reported among infants in the Pelotas cohort of Southern Brazil evaluated at 11 years of age. Compared to infants who received <1 m of BF, those who received 1 to <3 m and 3 to <6 m had a lower BMI at 11 years [6]. However, confidence intervals for these relations contained the null value and, thus, may have been due to chance. The lack of statistical significance in the Pelotas study may be because BF was defined as any BF, as opposed to exclusive BF as in the current study. That said, the Pelotas study similarly did not observe a clear dose–response relation, with the highest BMI observed among those who received the most BF (≥6 m). Another study conducted in Finland with the Helsinki Birth Cohort (offspring born between 1934 and 1944) reported a U-shape association between BF and later BMI. Specifically, individuals with a shorter duration of human milk feed (<2 m) and those with longer duration (≥8 m) had a higher BMI in adulthood and those with between 3 and 4 m had lower BMI compared to those who received between 5 and 7 m of BF [20]. In the current study, the protection of BF for later BMI was similarly not linear and did not continue beyond 5 years of age. In fact, at later ages (10, 16, and 23), in partial concordance with the results of the Helsinki study, the group that received the most exclusive BF (≥6 m) seemed to have higher BMI compared to those with the least BF, although the relationship was not statistically significant. While some trends seemed in accordance, exact comparisons between our findings and those of the Helsinki cohort should be made with caution, as many differences likely exist between the two. Further studies of long-term relations between BF and BMI, especially studies that can delineate duration of BF, are warranted.

The same article that reported results from the Pelotas cohort found a lower BMI at 7 years of age associated with longer BF among participants of the ALSPAC cohort from England. The authors concluded that the potential protective effect of BF with later BMI is not causal, but rather the result of residual confounding that is more common in high-income settings [6]. That is, BF habits may be more likely to be socially determined in high-income settings. One systematic review that attempted to compare results of hypothesized relations between BF and weight, height, and BMI in early life between developed and developing countries found that the strongest and most consistent inverse relations (i.e., more BF, lower BMI) came from the developed world (e.g., Australia [4] and the Netherlands [21]), with inconsistent findings from developing countries such as South Africa [22]. Although Chile is now considered a high-income and developed country, it was categorized as middle-income and developing when participants of the current study were infants and throughout their childhood. In fact, in Chile, the 1990s are often used as a point of comparison for several markers of development. For example, poverty was 38.6% in 1990 and was reduced to 13.7% by 2002 [23]. That said, it is important to mention that while the 1990s marked the beginning of an economic transition for Chile, the same was not true for social and educational changes that came later [14,15]. It is possible that this explains our somewhat unexpected results of a lack of clear relation between more and longer BF and lower BMI. Furthermore, there is evidence indicating some social patterning of BF habits in Chile. One government report summarizing information from prevalence studies of BF conducted between 1992 and 2002 shows a clear difference in BF duration by whether a woman works outside of or within the home [24]. A more recent study, conducted in 2011–2012 reported similar differences in duration of exclusive BF by years of education among women attending preventive appointments in the public health system [11]. Our sample does not include the full range of SES; families were recruited from low- to middle-income neighborhoods in the southeastern area of the capital [16]. While other analyses conducted in this sample have been able to appreciate changes in SES levels over time [25], most families were similar with respect to resources available during infancy. Had we had the full range of SES we may have found a clearer dose–response relation.

One potential mechanism proposed to explain how BF influences later health is the composition of breastmilk. Breastmilk contains a complex community of bacteria [26] that is nutrient rich. Another mechanism proposed is that BF is more “baby-led”, which facilitates a healthy satiety response that persists as an infant develops [27,28]. While BF is baby-led, even in the presence of continued BF, bottle feeding and the habits associated with the behavior may disrupt satiety response mechanisms. For example, caregivers, whether consciously or not, may encourage finishing of a bottle regardless of satiety, which has been associated with maternal encouraging of older children to finish all food on the plate [29]. Thus, we might expect a mixed or partial BF group to be more similar to a no BF group in terms of relations with later BMI, as both groups introduced bottles before those in the exclusive BF group. For this reason, we tested for a linear trend, as we would expect that as the amount of BF increased (exclusive BF > partial BF > no BF), so would the protection for later BMI. We found no evidence of a dose–response relation, but found that partial BF, but not exclusive BF at 6 m, related to lower BMI. The protective effects of more, versus less BF, we observed were only at the youngest ages. Thus, our results only partially support these hypotheses, although additional studies, potentially with larger sample sizes, are needed for further validation.

The results of this study should be interpreted considering its limitations. There was no data on age at introduction of other liquids (e.g., juice or water) or solid food. While we know that pureed fruits were often introduced after 4 m [30], the specific timing of introduction was not available. Thus, testing whether the introduction of solid foods or other liquids might have affected this study’s findings. While our analyses adjusted for some important variables known to influence BF and BMI, including infant sex, maternal education, gestational age, and birth weight, we were unable to adjust for other important factors. For example, we lacked information on maternal health during pregnancy, pregnancy weight gain, and mothers’ pre- and post-natal BMI, all of which represent potential unmeasured confounding variables. Maternal obesity in particular has been shown to influence the composition of breastmilk, which in turn may relate to early infant growth trajectory [31,32]. Taken together, it is possible that these two limitations explain the somewhat unexpected findings of our work. Another consideration is how exclusive BF was categorized. A study conducted in Canada comparing, in part, medical records with maternal reports showed that among infants categorized by mothers to have been “exclusively BF” many received small amounts of supplementary milk in the hospital [33]. The study further showed that this group had slightly higher BMI z-scores at 1 year compared to infants who were BF exclusively with no hospital supplementation. In Chile, the practice of providing supplementary milk during the short hospital stay is widespread, and often, as in Canada, parents are not even aware that it happens. Future studies should attempt to ascertain this level of information. Future studies might also consider evaluating the connections between BF and additional anthropometric indicators, as BMI does not differentiate between fat and muscle mass and BMI z-scores might be particularly problematic during certain developmental periods [34].

Some strengths should also be noted. Our study was conducted in Santiago, Chile at a time when Chile was considered a developing country. Given the possibility previously proposed of residual confounding explaining the relation between longer BF and lower BMI later in life, it is important to evaluate evidence from a variety of contexts. Furthermore, the measurement of duration of BF and introduction of the first bottle (equivalent, in the current study, to the end of exclusive BF), allowed us to differentiate between exclusive BF, no BF, and partial BF (combined bottle and BF). In addition, while recall bias is still possible, it was minimized as mothers were asked about feeding habits (continued BF and date of first bottle) on a weekly basis starting at 6 months of age and continuing until 1 year. Thus, in our study, if a mother provided the first bottle to her infant at 4 months of age and continued BF, she would only have to remember 2 months previous. Finally, the ability to evaluate relations at multiple time-points in infancy, childhood, adolescence, and young adulthood was a strength as it is possible that longer BF may relate to lower BMI at younger ages only, as was the case in the current study.

## 5. Conclusions

In conclusion, the study findings provide limited evidence of a protective effect of longer BF for lower BMI in childhood. While we observed some statistically significant results implying lower BMI z-scores at 1 and 5 y among infants who were partially BF, compared to those who were not BF at 6 m, and those who were BF for between 3 and 6 months, compared to <1 m, given the magnitude of the effect and the lack of a clear dose–response relationship, we caution against over-interpretation. The current results from low- to middle-income Chilean infants are in partial agreement with other studies conducted in developing countries and imply that BF may not be causally related to later BMI. Our findings may reflect the importance of other, unmeasured variables that influence later BMI (e.g., child physical activity, maternal characteristics, social environment). Despite the lack of evidence of a clear protective effect of BF for lower BMI, the benefits of BF for other child and maternal health outcomes in both the short- and long-term are unequivocal and BF should continue to be encouraged for all.

## Figures and Tables

**Table 1 nutrients-15-03121-t001:** Descriptive statistics of a subset of participants from the Santiago Longitudinal Study (*n* = 821): 1994–2017.

	Missing	Mean (SD)
*Parental characteristics*		
Maternal age, years	0	26.5 (6.1)
Maternal highest education completed, years	0	9.5 (2.6)
*Infancy characteristics*		
Male sex	0	436 (53.1%)
Birth weight	4	3.5 (0.3)
Gestational age	4	39.4 (1.0)
*Type of BF at 6 months*	11	
No		218 (26.9%)
Partial		283 (34.9%)
Exclusive		309 (38.1%)
*Duration of exclusive BF, months*	6	
<1		134 (16.4%)
1 to <3		130 (16.0%)
3 to <6		242 (29.7%)
≥6		309 (37.9%)
*BMI z-score, years*		
1	5	0.8 (0.9)
5	283	1.0 (1.1)
10	287	1.0 (1.1)
16	313	0.9 (1.1)
23 ^1^	310	26.7 (5.5)

^1^ BMI = kg/m^2^; BF = breastfeeding.

**Table 2 nutrients-15-03121-t002:** BMI at 1, 5, 10, 16, and 23 years in early childhood (1 year), childhood (5 and 10 years), adolescence (16 years), and young adulthood (23 years) by BF type and duration in a sample of Chilean participants ^1^.

	BMI z-Score				BMI
	1 *n* = 805	5 *n* = 531	10 *n* = 526	16 *n* = 501	23 *n* = 466
Type of BF at 6 months					
No	0.91 (0.87)	1.11 (1.19)	1.01 (1.21)	0.83 (1.20)	26.2 (5.45)
Partial	0.72 (0.96)	0.85 (1.17)	0.96 (1.15)	0.85 (1.15)	26.3 (5.14)
Exclusive	0.84 (0.96)	1.11 (1.17)	1.20 (1.11)	1.05 (1.03)	27.2 (5.03)
Duration of exclusive BF, months					
<1	0.92 (0.83)	1.14 (1.37)	1.14 (1.21)	1.01 (1.13)	26.9 (6.37)
1 to <3	0.91 (0.92)	0.98 (1.16)	0.98 (1.08)	0.86 (1.09)	26.5 (4.95)
3 to <6	0.69 (0.96)	0.83 (1.05)	0.90 (1.20)	0.75 (1.24)	25.8 (4.70)
≥6	0.84 (0.96)	1.10 (1.17)	1.20 (1.11)	1.06 (1.03)	27.2 (5.03)

^1^ Unadjusted mean (SD); BF = breastfeeding.

**Table 3 nutrients-15-03121-t003:** Adjusted linear regression models ^1,2^ of the relationships between BF (type and duration) and BMI in different years of childhood, adolescence, and young adulthood in a sub-set of the Santiago Longitudinal Study.

	BMI z-Score				BMI
	1	5	10	16	23
Type of BF at 6 months (No BF, reference group)					
Partial	−0.25 −0.41, 0.08	**−0.31** **−0.57, −0.05**	−0.07 −0.33, 0.18	<0.01 −0.25, 0.25	0.10 −1.10, 1.30
Exclusive	−0.12 −0.28, 0.05	−0.04 −0.29, 0.22	0.18 −0.07, 0.43	0.22 −0.04, 0.47	1.00 −0.19, 2.18
*p*-value of linear trend	0.27	0.93	0.11	0.07	0.13
Duration of exclusive BF, month (<1 month, reference group)					
1 to <3	0.05 −0.17, 0.27	−0.13 −0.49, 0.22	−0.12 −0.46, 0.23	−0.12 −0.46, 0.22	−0.29 −1.95, 1.37
3 to <6	**−0.22** **−0.41, −0.02**	**−0.31** **−0.61, −0.01**	−0.22 −0.52, 0.07	−0.26 −0.55, 0.04	−1.10 −2.50, 0.30
≥6	−0.07 −0.25, 0.12	−0.03 −0.32, 0.25	0.09 −0.20, 0.37	0.06 −0.22, 0.01	0.30 −1.05, 1.66
*p*-value of linear trend	0.22	0.91	0.43	0.58	0.92

^1^ Adjusted for child sex, maternal education, maternal age, gestational age, and birth weight. ^2^ Values shown are unstandardized coefficients (95% confidence intervals) unless otherwise stated. Bolded values represent *p* < 0.05.

## Data Availability

The data presented in this study are available in the article.

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
