# Peer review of "Breastfeeding Duration and Timing of Bottle Supplementation: Associations with Body Mass Index from Childhood to Young-Adulthood"

_nutrients, 2023, doi:10.3390/nu15143121_

Round 1

Reviewer 1 Report

The article is well written and clear. The topic is interesting. The study is well conducted.

Given that the effectiveness of the BMI z-score is still a matter of debate (for istance, see Peterson et al. JAMA Pediatr. 2017;171(7):629-636. doi: 10.1001/jamapediatrics.2017.0460. ), I would suggest considering multiple measures in assessing obesity (e.g. anthropometric data, impedancemetry, or x-ray axial densitometry) and other factors (such as family lifestyle: diet, physical activity, family history of obesity, ...) in future studies.

Author Response

Below we address all specific questions raised in the review (italics) and detail how the manuscript was revised in response to suggestions. 

Reviewer 1:

  1. The article is well written and clear. The topic is interesting. The study is well conducted.

We appreciate that the Reviewer found the topic interesting and found that our study was well conducted.

  1. Given that the effectiveness of the BMI z-score is still a matter of debate (for istance, see Peterson et al. JAMA Pediatr. 2017;171(7):629-636. doi: 10.1001/jamapediatrics.2017.0460. ), I would suggest considering multiple measures in assessing obesity (e.g. anthropometric data, impedancemetry, or x-ray axial densitometry) and other factors (such as family lifestyle: diet, physical activity, family history of obesity, ...) in future studies.

We absolutely agree with the Reviewer that the use of BMI and BMI z-score is limited. We have included this limitation in the discussion section and have added the article provided to further justify this point (see lines 297-301).

Reviewer 2 Report

In this manuscript, the authors perform the project based on big clinical data numbers. By utilizing a more specific categorized method of non-/partial-/full- breastmilk feed, the authors provide us valuable data regarding the relationship between early breastmilk feed and later body mass index from a developing country society (the time when this study was conducted). And here are some comments:

In page 1 of 10, line 19-22, the author should explain the m and y more clearly, which are first time introduced in the manuscript (I assume m is month and y is year).

Is there a typo issue of the sentence “…given that the exposure variable (BF) is often defined differently” (page 2 of 10, line 76)?

Even already labeled later in Table 1, I think it will be good to mention the data here “3.58 kg (0.3) at birth, with slightly more males (53%) than females in the cohort. Mothers were on average 26.5 (6.1) years of age, with 9.5 (2.6) years of education” (page 3 of 10, line 14-146) are mean (SD).

In Table 2, after 1 year measurement, the data number reduced by ~30% start on 5 year and remain relatively unchanged till end of the research. I suggest the authors give an explanation of why there is a big reduction in data numbers. Also, as there is no significant difference in Table 2 (as the authors mentioned at page 4 of 10, line160-161), I suggest to re-write the sentence “Additionally, the exclusive BF group had a lower BMI z-score than those with no BF at 1 year but a higher BMI at 10, 16 and 23 years” (page 4 of 10, line156-157), because when you use words “lower/higher…than…”, that statement must based on significant difference.

In Table 3, compare to the reference group, each group at specific time point shows three data readings, the authors can give a short explanation to introduce what these three data represent.

The authors have a good discussion section with well covered factors that may affect the outcomes of the breastmilk feed vs. later body mass. And here are some minor suggestions:

Considering the data in Table 3 were adjusted by five factors (page 5 of 10, line 179), the authors can discuss the necessity and sufficiency of using these factors for adjustment.

In discussion section, the authors mentioned a study that was done in Finland, please discuss the compatibility of that study (1934-1944 in Helsinki) with this current manuscript (1994-2017 in Santiago).

Author Response

Below we address all specific questions raised by the reviewer (italics) and detail how the manuscript was revised in response to suggestions (regular text). 

Reviewer 2:

In this manuscript, the authors perform the project based on big clinical data numbers. By utilizing a more specific categorized method of non-/partial-/full- breastmilk feed, the authors provide us valuable data regarding the relationship between early breastmilk feed and later body mass index from a developing country society (the time when this study was conducted). And here are some comments:

  1. In page 1 of 10, line 19-22, the author should explain the m and y more clearly, which are first time introduced in the manuscript (I assume m is month and y is year).

Corrected.

Is there a typo issue of the sentence “…given that the exposure variable (BF) is often defined differently” (page 2 of 10, line 76)?

We have revised this sentence.

Even already labeled later in Table 1, I think it will be good to mention the data here “3.58 kg (0.3) at birth, with slightly more males (53%) than females in the cohort. Mothers were on average 26.5 (6.1) years of age, with 9.5 (2.6) years of education” (page 3 of 10, line 14-146) are mean (SD).

Corrected.

In Table 2, after 1 year measurement, the data number reduced by ~30% start on 5 year and remain relatively unchanged till end of the research. I suggest the authors give an explanation of why there is a big reduction in data numbers.

We apologize that this was not clear in the original article. In the methods section, we have clarified that the reduction was due to a substantial budget cut in the funding for the 5-year follow-up in which only a subset of a cohort could be followed (lines 116-18). Although the full cohort was invited to participate in subsequent evaluations (10, 16, and 23 years), as noted by the Reviewer, the sample size was relatively stable after the reduction in the portion of the cohort evaluated in the current analysis.

Also, as there is no significant difference in Table 2 (as the authors mentioned at page 4 of 10, line160-161), I suggest to re-write the sentence “Additionally, the exclusive BF group had a lower BMI z-score than those with no BF at 1 year but a higher BMI at 10, 16 and 23 years” (page 4 of 10, line156-157), because when you use words “lower/higher…than…”, that statement must based on significant difference.

We have modified the wording in this paragraph to make it clear that no comparisons were statistically significant and that only descriptive, and not statistical, comparisons are being made.

In Table 3, compare to the reference group, each group at specific time point shows three data readings, the authors can give a short explanation to introduce what these three data represent.

We appreciate this comment and have added 2 sentences in the paragraph describing the results of Table 3 to better explain and interpret results.

The authors have a good discussion section with well covered factors that may affect the outcomes of the breastmilk feed vs. later body mass. And here are some minor suggestions:

Considering the data in Table 3 were adjusted by five factors (page 5 of 10, line 179), the authors can discuss the necessity and sufficiency of using these factors for adjustment.

We appreciate this important comment. We have revised our limitations section to better present the necessity and sufficiency of the variables we controlled for in the analyses. See Lines 275-283.

In discussion section, the authors mentioned a study that was done in Finland, please discuss the compatibility of that study (1934-1944 in Helsinki) with this current manuscript (1994-2017 in Santiago).

This is an important point. We have added a sentence to caution over-interpretation of comparisons.

Reviewer 3 Report

In this manuscript entitled "Breastfeeding duration and timing of bottle supplementation: association with body mass index from childhood to young-adulthood.”, the authors tested the hypothesis that the greater the amount of BF, the greater the preventive effect on subsequent BMI. As a result, this was only observed in younger age groups. Therefore, this hypothesis is only partially supported, but the results are very interesting.

Comments:

1.    As the authors mention, the significance of conducting this study in Chile, which has been developing since the 1990s, should be written in the Introduction.

2.    What are the reasons or mechanisms why partial or mixed nutrition is thought to increase BMI? The authors should mention this in comparison to the components in breast milk in Discussion.

3.    Is it possible that a decrease in BMI, especially at a younger age, could affect BMI at later ages? How the body's metabolism, etc. is affected should be discussed.

Author Response

Below we address all specific questions raised by the reviewer (italics) and detail how the manuscript was revised in response to suggestions (regular text). 

Reviewer 3:

In this manuscript entitled "Breastfeeding duration and timing of bottle supplementation: association with body mass index from childhood to young-adulthood.”, the authors tested the hypothesis that the greater the amount of BF, the greater the preventive effect on subsequent BMI. As a result, this was only observed in younger age groups. Therefore, this hypothesis is only partially supported, but the results are very interesting.

As the authors mention, the significance of conducting this study in Chile, which has been developing since the 1990s, should be written in the Introduction.

We thank the Reviewer for this suggestion. Revised as suggested.

What are the reasons or mechanisms why partial or mixed nutrition is thought to increase BMI? The authors should mention this in comparison to the components in breast milk in Discussion.

We appreciate this very interesting comment and suggestions for how and where to incorporate this discussion. Please see lines 271-278.

Is it possible that a decrease in BMI, especially at a younger age, could affect BMI at later ages? How the body's metabolism, etc. is affected should be discussed.

The Reviewer brings up an interesting point. In our birth cohort, while we certainly can identify some individual cases of decreases in BMI overtime, overall, we observed that average BMI remained relatively stable (Table 1). Given the context of our results, a discussion of how decreases in BMI affect BMI at later ages appears to be outside of the scope of the current study.

Reviewer 4 Report

The authors provide a very interesting research study for the association between breastfeeding duration and later body mass index. The introduction, method, results and discussion are clearly presented, and in a manner that is easily understandable. However, there are a few concerns that they need to take care of:

The English language and grammar needs to be checked up.

At line 81, the phrase needs to be re-written,  to make sense.

The Introduction section doesn't contain a concise and clear conclusion. It is very important that the authors focus on the main reason, scope and goal of the manuscript. Also, they should emphasis on the novelty of the work, future possibilities and so on.

At line 141, the explanation for SPSS is missing. 

The conclusions needs to be written in a manner that reflects the study conducted by the authors. No future perspectives are provided, nor any possible encounters they may face in the future regarding the method they used.

The English language and grammar needs to be checked up.

Author Response

We very much appreciate the reviewer´s critique of our manuscript. Below we address all specific questions raised by the reviewer (italics) and detail how the manuscript was revised in response to suggestions (regular text). We believe these revisions strengthen the clarity and impact of the manuscript. 

Reviewer 4:

The authors provide a very interesting research study for the association between breastfeeding duration and later body mass index. The introduction, method, results and discussion are clearly presented, and in a manner that is easily understandable. However, there are a few concerns that they need to take care of:

The English language and grammar needs to be checked up.

Five native English speakers have carefully reviewed our manuscript for language and grammar concerns.

At line 81, the phrase needs to be re-written,  to make sense.

Revised.

The Introduction section doesn't contain a concise and clear conclusion. It is very important that the authors focus on the main reason, scope and goal of the manuscript. Also, they should emphasis on the novelty of the work, future possibilities and so on.

We thank the Reviewer for this thoughtful comment. We have revised the introduction section, in particular, the final two paragraphs (lines 75-101) to emphasize the justification for our study, the novelty of our work, and how the results will contribute to the scientific evidence related to breastfeeding and later BMI.  

At line 141, the explanation for SPSS is missing. 

This abbreviation has been defined in the revised manuscript.

The conclusions needs to be written in a manner that reflects the study conducted by the authors. No future perspectives are provided, nor any possible encounters they may face in the future regarding the method they used.

We have added a sentence summarizing the main findings of the current study. Ideas for future research, including ideas related to the methodology used, are provided throughout the entire discussion (see lines 235-7, 266-7, 275-6, 277-301).